# Rethinking Perceptual Metrics for Medical Image Translation

**Nicholas Konz**[1]                                          NICHOLAS.KONZ@DUKE.EDU
**Yuwen Chen**[1]                                               YUWEN.CHEN@DUKE.EDU
**Hanxue Gu**[1]                                                 HANXUE.GU@DUKE.EDU
**Haoyu Dong**[1]                                           HAOYU.DONG151@DUKE.EDU
**Maciej A. Mazurowski**[1,2,3,4]                       MACIEJ.MAZUROWSKI@DUKE.EDU
[1]*Dept. of Electrical and Computer Engineering,* [2]*Dept. of Radiology,* [3]*Dept. of Computer Science,*
[4]*Dept. of Biostatistics & Bioinformatics, Duke University, NC, USA*

**Editors:** Accepted for publication at MIDL 2024

## Abstract

Modern medical image translation methods use generative models for tasks such as the conversion of CT images to MRI. Evaluating these methods typically relies on some chosen downstream task in the target domain, such as segmentation. On the other hand, task-*agnostic* metrics are attractive, such as the network feature-based perceptual metrics (*e.g.*, FID) that are common to image translation in general computer vision. In this paper, we investigate evaluation metrics for medical image translation on two medical image translation tasks (GE breast MRI to Siemens breast MRI and lumbar spine MRI to CT), tested on various state-of-the-art translation methods. We show that perceptual metrics do not generally correlate with segmentation metrics due to them extending poorly to the anatomical constraints of this sub-field, with FID being especially inconsistent. However, we find that the lesser-used *pixel-level* SWD metric may be useful for subtle intra-modality translation. Our results demonstrate the need for further research into helpful metrics for medical image translation.

**Keywords:** image translation, evaluation metrics, breast MRI, MRI-to-CT

## 1. Introduction

Medical image translation is the task of altering a medical image to look like it was taken in a different scanner or modality, and it has requirements that differ from mainstream natural image translation/style transfer. Namely, it is crucial to preserve anatomical/semantic content during translation. While natural image translation papers often employ perceptual learned-feature metrics like FID (Heusel et al., 2017), these metrics are not suitable for medical images because they may fail to capture global anatomical consistency and realism, as demonstrated previously (Jayasumana et al., 2024; Chen et al., 2024; Konz et al., 2024) and in this work. A common metric that is more anatomy-focused is the performance of a segmentation model trained in the target domain on translated images (Vorontsov et al., 2022; Kang et al., 2023) (or the reverse (Yang et al., 2019)). Yet, this not only requires annotations for training and evaluating the segmentation model, but it may also be overly reductionist, leading to biased evaluations for specific tasks. In reality, translated images could be used for many applications, urging us to investigate better usage, understanding, and future development of *downstream task-agnostic* metrics, *e.g.*, perceptual metrics.

## 2. Methods

We evaluate two medical image translation tasks: (1) subtle *intra*-modality breast MRI translation and (2) a more drastic *inter*-modality translation of lumbar spine MRI to CT.

**Datasets and Preprocessing.** We use the public DBC dataset (Saha et al., 2018) for our dataset of 2D pre-contrast breast MRI slice images, following the same subsetting, preprocessing, and splits of (Konz et al., 2024), with accompanying breast and fibroglandular tissue (FGT) segmentations from (Lew et al., 2024). Each split contains scans from both Siemens-manufactured (source domain) and GE-manufactured scanners (target domain). For lumbar spine MRI-to-CT slice translation, we follow the same preprocessing and train/validation/test splits of (Chen et al., 2024) which did MRI-to-CT translation, using the same private lumbar spine MRIs for the source domain, and lumbar spine CTs from the public TotalSegmentator dataset (Wasserthal et al., 2023) for the target domain, both of which are accompanied by bone segmentations. All images are resized to $256 \times 256$ and normalized linearly to $[0, 255]$. Altogether, this forms train/validation/test splits of data from the source and target domains of $\{4096, 7900\}/\{432, 1978\}/\{688, 1890\}$ images for breast MRI, respectively, and $\{495, 1466\}/\{175, 409\}/\{158, 458\}$ for MRI-to-CT.

**Translation Models.** We evaluate unpaired image translation models of CycleGAN (Zhu et al., 2017), MaskGAN (Phan et al., 2023), UNSB (Kim et al., 2024), and SPADE (Park et al., 2019). Unlike the other models, SPADE is an anatomically-guided segmentation-conditional model (denoted as "SPADE†"), trained on target domain images *and segmentation* pairs to generate translations from source domain image segmentations. All models were trained on target and source domain images (or only target domain images for SPADE) from the training set and evaluated using source domain images from the test set.

**Segmentation Metrics for Image Translation.** We first evaluate the metric of (1) training and validating a UNet on the corresponding target domain training set to segment the aforementioned object(s) of interest, and (2) evaluating it on test set images translated from the source domain with respect to their original segmentation labels, via the average prediction Dice coefficient.

**Perceptual Metrics for Image Translation.** We also assess perceptual metrics that compare translated image quality to real target domain images. These include FID (Heusel et al., 2017), the distance between Gaussian feature distributions extracted by an ImageNet-pretrained Inception network; KID (Bińkowski et al., 2018),

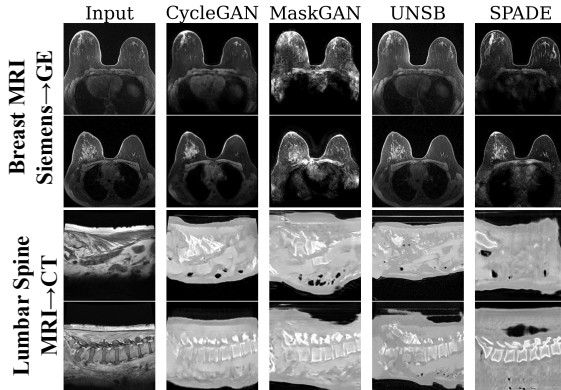

Figure 1: Example translations for each model.

like FID but without the Gaussian assumption and better for small datasets; and IS (Salimans et al., 2016), which assesses feature diversity and quality. We also test the less common

SWD (Karras et al., 2018), which focuses on textural and pattern similarity by calculating the distance between **pixel-level**, not learned, features. In the case of having $N_{trans}$ translated images and $N_{tgt}$ target domain images to compare such that $N_{trans} \neq N_{tgt}$, we randomly sample $N = \min(N_{trans}, N_{tgt})$ images from each set to compute the distance metrics. We also note that FID may be inaccurate for our relatively small datasets ($N < 2048$, where the feature covariance matrix is not full-rank); noting this, we report it as "FID*".

## 3. Results and Discussion

In Table 1 we show the performance of all translation models according to all metrics. For the perceptual metrics, a lower value is better for distances (FID, KID, SWD), and a higher one is better for IS. We also show example translated images in Fig. 1, and the correlations of perceptual metrics with segmentation metrics are in Fig. 2.

| | Breast MRI Siemens→GE Translation | | | | | | Lumbar Spine MRI→CT Translation | | | | |
|---|---|---|---|---|---|---|---|---|---|---|---|
| | Dice (↑) | | Perceptual Metrics | | | | Dice (↑) | Perceptual Metrics | | | |
| **Method** | Breast | FGT | FID* | KID | SWD | IS | Bone | FID* | KID | SWD | IS |
| *None* | 0.927 | 0.651 | 144 | 0.069 | 705 | 2.58 | 0.007 | 323 | 0.300 | 1553 | **2.93** |
| CycleGAN | 0.934 | 0.529 | **107** | **0.049** | 556 | 2.73 | 0.229 | 210 | **0.161** | 960 | 2.29 |
| MaskGAN | 0.865 | 0.277 | 118 | 0.089 | 1037 | **3.00** | 0.158 | 248 | 0.217 | 1114 | 2.22 |
| UNSB | 0.934 | 0.646 | 156 | 0.079 | 756 | 2.46 | 0.138 | **208** | 0.172 | **932** | 2.14 |
| SPADE† | **0.950** | **0.707** | 119 | 0.067 | **500** | 2.91 | **0.942** | 251 | 0.242 | 1359 | 2.29 |

Table 1: Quantitative results for both translation tasks. Best and runner-up models are shown in bold and underlined according to each metric, respectively.

Overall, **perceptual metrics do not consistently align with common segmentation metrics for medical image translation.** No single perceptual metric reliably correlates with segmentation metrics for both breast MRI and MRI-to-CT translation. Using a perceptual metric for model selection will depend highly on the choice of metric, with the commonly used FID being especially inconsistent. Therefore, we advise caution in using FID for evaluating medical image translation.

SWD shows a better correlation than the learned feature metrics (FID, KID, IS) for the subtle intra-modality breast MRI translation. However, SWD fails for the more complex inter-modality translation of MRI-to-CT, likely due to its focus on pixel-level changes which are insufficient for capturing larger visual differences.

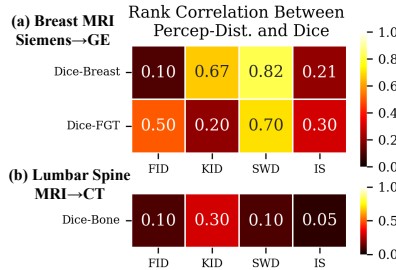

Figure 2: Absolute correlation of perceptual metrics with segmentation metrics.

Given that perceptual metrics are designed for assessing image realism rather than object preservation during translation, their limited correlation with segmentation metrics is understandable. Nonetheless, this indicates that perceptual metrics may not be fully suitable for medical image translation. A broader evaluation approach and research into more universally applicable metrics are needed in this field.

## Acknowledgements

Research reported in this publication was supported by the National Institute Of Biomedical Imaging And Bioengineering of the National Institutes of Health under Award Number R01EB031575. The content is solely the responsibility of the authors and does not necessarily represent the official views of the National Institutes of Health.

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
