# OpenReview forum: "Rethinking Perceptual Metrics for Medical Image Translation"
_MIDL.io/2024/Short_Papers — MIDL 2024 Short Papers_

### Official Review · Reviewer_skR9 · 2024-04-24

**Confidence:** 4
**Final Rating:** 4

**Review:**

Evaluation of medical image-to-image translation models is an open and challenging problem. The commonly used FID metric for natural images often fails for medical images due to not capturing anatomical consistency of medical images and also due to relying on features that were trained from a statistically very different domain. Downstream segmentation task as an evaluation offers an alternative but is costly due to requiring labels. This paper uses several metrics including the above on two medical image translation tasks, one subtle domain change and another more drastic translation from MRI to CT. Not surprisingly the paper confirms that the FID metric fails to correlate with the segmentation metric. The SWD metric, a pixel-level metic that does not use learned features, correlates with the segmentation metric for the subtle domain shift translation problem but fails for the more drastic one. These results could help offer guidance to researchers looking for ways to evaluate their image translation methods.

---

### Decision · Program_Chairs · 2024-04-26

Accept